# Not All Errors Are Made Equal: A Regret Metric for Detecting System-level Trajectory Prediction Failures

**Kensuke Nakamura**[1]    **Ran Tian**[2]    **Andrea Bajcsy**[1]
[1]Carnegie Mellon University    [2] UC Berkeley
{kensuken, abajcsy}@andrew.cmu.edu,  rantian@berkeley.edu

**Abstract:** Robot decision-making increasingly relies on data-driven human prediction models when operating around people. While these models are known to mispredict in out-of-distribution interactions, only a subset of prediction errors impact downstream robot performance. We propose characterizing such "system-level" prediction failures via the mathematical notion of regret: high-regret interactions are precisely those in which mispredictions degraded closed-loop robot performance. We further introduce a probabilistic generalization of regret that calibrates failure detection across disparate deployment contexts and renders regret compatible with reward-based and reward-free (e.g., generative) planners. In simulated autonomous driving interactions and social navigation interactions deployed on hardware, we showcase that our system-level failure metric can be used offline to automatically identify challenging closed-loop human-robot interactions that generative human predictors and robot planners previously struggled with. We further find that the very presence of high-regret data during predictor fine-tuning is highly predictive of closed-loop robot performance improvement. Additionally, fine-tuning with the informative but significantly smaller high-regret data (23% of deployment data) is competitive with fine-tuning on the full deployment dataset, indicating a promising avenue for efficiently mitigating system-level human-robot interaction failures. Project website: https://cmu-intentlab.github.io/not-all-errors/

**Keywords:** Human-Robot Interaction, Trajectory Prediction, Failure Detection

## 1   Introduction

From autonomous cars in cities [1, 2, 3] to tabletop manipulators at home [4, 5, 6], robots operating around people rely on data-driven trajectory during planning to to anticipate how others will behave. Despite their widespread adoption in robot autonomy pipelines, current human prediction models are still imperfect: they frequently mis-predict when faced with out-of-distribution interactions [7, 8] and are prone to causal misidentification [9] wherein the model "lazily" converges to incorrect correlates in the data. These prediction errors can lead to critical downstream robot system failures, impeding performance at best and compromising safety at worst. Thus, analyzing when and how trajectory prediction errors lead to robot failures is critical for developing runtime monitors, creating safety-focused benchmarks, and incrementally improving the prediction models themselves.

While current methods detect interaction data that is dissimilar to the training distribution [10] or leads to highly uncertain or erroneous predictions [11, 12, 13], these are all fundamentally *component-level* approaches that detect only when the predictor is faulty in isolation. However, prediction errors at the component-level do not always lead to *system-level* robot decision-making errors downstream [14]. Consequently, relying on component-level failures to evaluate, monitor, or improve the prediction model is inaccurate and inefficient. For example, consider Figure 1. Even though the robot's predictions were faulty in all three scenarios, it is only in the latter two where this

8th Conference on Robot Learning (CoRL 2024), Munich, Germany.

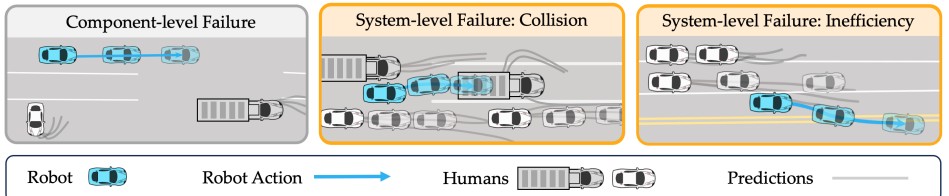

Figure 1: All scenarios have component-level prediction failures: mispredicting that parked cars will turn (left), marooned truck will to move (center), and nearby cars will lane change (right). But, only the center and right scenarios have system-level prediction failures which impact robot performance.

caused undesirable closed-loop robot behavior: collisions in the center and inefficiency in the right. The core challenge we address is the automatic detection of such system-level prediction failures.

Our key insight is that **the mathematical notion of regret is a rigorous way to identify system-level prediction failures.** High-regret interactions are precisely those in which mispredictions caused the robot to make a suboptimal decision in hindsight. In Section 4 we formalize this idea and introduce a probabilistic generalization of regret that no longer depends on explicit reward functions for regret computation, extending its applicability to reward-free planners such as generative models. In Section 6, we extract system-level failures offline from a dataset of collected deployment interactions and compare our approach to alternative failure detection methods with a state-of-the-art autonomous driving trajectory predictor and planner, and demonstrate the compatibility of our approach with generative neural planners in a social navigation setting deployed on hardware with on-board sensing. Finally, in Section 7, we conduct a case study where we find that fine-tuning a robot's human behavior predictor on the small amount of high-regret data (23% of the deployment data) is highly predictive of system-level performance improvements at re-deployment.

## 2 Related Work

**Interaction-aware Prediction.** Interactive robots commonly use conditional behavior predictions to model the influence between the robot's decisions and other agents' future behavior [15, 16, 4, 17]. More recently, there has also been a shift towards using multi-agent prediction models or generative "foundation models" as planners [18, 19, 20, 21, 22, 23, 24]. However, these models are overwhelmingly evaluated on log-replay data, missing out on the closed-loop interaction effects between agents. This may lead to *overly optimistic* estimations of how the robot may influence the behavior of others [25] and poorly affect downstream performance. We introduce a system-level metric for evaluating interaction-aware predictors that quantifies closed-loop performance of the trajectory predictor. We also show that fine-tuning such a predictor on identified system-level failures leads to significant downstream performance improvements.

**Component-level Failure Detection.** Many prior works have studied the detection of anomalous or out-of-distribution data (see [26] and [27] for surveys) relative to a prediction model's training data: classifying poor predictions based on average displacement error (ADE) [12, 28], Bayesian filtering [8, 29, 30], kernel density estimation [31], uncertainty [11, 32], conformal prediction [33], and evidential deep learning [10]. However, these approaches are fundamentally component-level, ignoring the "downstream" effects of errors in the predictor on robot decision-making.

**System-level Prediction Metrics.** Our work contributes to the small-yet-growing area of developing system-level metrics for evaluating components of the autonomy stack [34, 14, 35, 34, 36, 37, 38]. Existing system-level trajectory prediction metrics compare the difference between predicted and incurred costs [37] or weight prediction errors by a task-dependent cost function [38, 39, 40]. In addition to evaluating predictors, system-level metrics can also be useful in the offline setting, wherein datasets are mined for challenging examples. For instance, Stoler et al. [41] mine for safety-critical interactions via heuristic features and Hsu et al. [42] mine via counterfactual responsibility metrics. Our approach is also instantiated in an offline setting, but utilizes our novel regret metric to identify system-level human-robot interaction failures. Furthermore, our method does not require a cost function to detect system-level failures, rendering our method compatible with reward-free planners.

## 3    Problem Formulation

**Agent States, Actions, and Context.** Let the robot state be denoted by $s^{\mathrm{R}} \in \mathbb{R}^{n_R}$, and the state of the $M$ human agents in the scene be denoted by $s^{\mathrm{H}_i} \in \mathbb{R}^{n_i}$, $i \in \{1, \ldots, M\}$. For example, for an autonomous driving application domain, each agent's state can be represented as position, heading, and velocity. All agents' states evolve according to their respective control actions: $a^{\mathrm{R}} \in \mathbb{R}^{m_R}$ for the robot (e.g., acceleration and turning rate) and $a^{\mathrm{H}_i} \in \mathbb{R}^{m_i}$ for the $M$ humans, where $n_R$ and $m_R$ are the dimensions of the robot state and action and are similarly defined for each human. For shorthand, we denote the joint human-robot state at any time $t$ as $s_t = [s_t^{\mathrm{R}}, s_t^{\mathrm{H}_1}, \ldots, s_t^{\mathrm{H}_M}]$. Let a $T$-length future sequence of actions starting at time $t$ be $\boldsymbol{a}_t^{\mathrm{H}_i} = [a_t^{\mathrm{H}_i}, \ldots, a_{t+T}^{\mathrm{H}_i}]$ any human $i$ and $\boldsymbol{a}_t^{\mathrm{R}} = [a_t^{\mathrm{R}}, \ldots, a_{t+T}^{\mathrm{R}}]$ for the robot. Finally, let $C$ denote the scene or environment context (e.g., semantic map of the environment) used for both human prediction and robot decision-making.

**Human Prediction & Robot Planning.** We assume the robot has access to a pre-trained human behavior prediction model $P_\theta$ with parameters $\theta$ capable of conditional generations [16, 18]. Mathematically, this predictor is:

$$P_\theta(\boldsymbol{a}_t^{\mathrm{H}_1:\mathrm{H}_M} \mid \boldsymbol{a}_t^{\mathrm{R}}, s_t, C). \tag{1}$$

The model can generate $T$-horizon predictions of all $M$ humans in the scene, $\boldsymbol{a}_t^{\mathrm{H}_1:\mathrm{H}_M} := [\boldsymbol{a}_t^{\mathrm{H}_1}, ..., \boldsymbol{a}_t^{\mathrm{H}_M}]$, conditioned on a robot action trajectory $\boldsymbol{a}_t^{\mathrm{R}}$, scene context $C$, and current[1] human-robot state $s_t$. We also use the robot's task-driven planner:

$$\boldsymbol{a}_t^{\mathrm{R}} = \pi_\phi(s_t, P_\theta, C), \tag{2}$$

with parameters $\phi$. This planner generates a best-effort robot plan $\boldsymbol{a}_t^{\mathrm{R}}$ with respect to the current joint state $s_t$, scene context $C$ and predictions, $P_\theta$. Within our framework, the robot's planner could be an optimization-based planner that uses a reward function (where $\phi$ are feature weights of a pre-specified reward function) or a neural planner (where $\phi$ are the pre-trained neural network weights). At deployment time, the robot's planning model can be re-queried given new observations and predictions in a receding horizon fashion. We assume that the planner is optimal under the given reward function or the available data for planner design.

**Problem Statement.** We deploy the robot for $\hat{T}$ time steps in $N$ scenarios (e.g., different initial conditions, environments) and collect an aggregate deployment dataset $\mathcal{D} = \{(\hat{\boldsymbol{s}}, \hat{\boldsymbol{a}}^{\mathrm{R}}, \hat{\boldsymbol{a}}^{\mathrm{H}_1:\mathrm{H}_M})_n\}_{n=1}^N$ consisting of observed state and action trajectories for the robot and $M$ humans. Importantly, this observed data exhibits how real interactions between the humans and robot occur, and are a function of the robot's predictive human model used during planning. We seek to automatically identify a subset of deployment data, $\mathcal{D}_{\mathcal{F}} \subset \mathcal{D}$, that captures *system-level prediction failures*: deployment interactions wherein prediction failures significantly impacted *closed-loop* robot performance.

## 4    A Regret Metric for Detecting System-level Trajectory Prediction Failures

Canonically, regret measures the reward ($\mathcal{R}_\phi$) difference between the optimal action the robot could have taken *in hindsight* ($\boldsymbol{a}^{\mathrm{R}}$) and the executed action ($\hat{\boldsymbol{a}}^{\mathrm{R}}$) the robot took under uncertainty [43]: $\mathrm{Reg}_t(d) := \max_{\boldsymbol{a}_t^{\mathrm{R}}} \mathcal{R}_\phi(\boldsymbol{a}_t^{\mathrm{R}}, \hat{\boldsymbol{a}}_t^{\mathrm{H}_1:\mathrm{H}_M}, \hat{\boldsymbol{s}}_t, C) - \mathcal{R}_\phi(\hat{\boldsymbol{a}}_t^{\mathrm{R}}, \hat{\boldsymbol{a}}_t^{\mathrm{H}_1:\mathrm{H}_M}, \hat{\boldsymbol{s}}_t, C)$, $d \in \mathcal{D}$. If we were to rank all the interactions in $\mathcal{D}$ by the robot's regret, we could identify those with the highest regret as a means of building our system-level failure dataset, $\mathcal{D}_{\mathcal{F}}$. While regret is a principled measure of system-level failures, directly using rewards during regret computation may not always be desirable. First, not all robot planners use reward functions—for example, generative planners fit joint distributions to multi-agent behavior [18] and choose trajectories via their likelihood rather than an explicit reward. This immediately makes canonical regret impossible to evaluate. Second, the canonical regret definition assumes that the same difference in reward values in different deployment contexts are fairly comparable. Unfortunately, in practice, when robots are deployed in a wide variety of scenarios and contexts $C$, it is difficult to have such a perfectly calibrated reward. We illustrate this technical point with an example from our simulation experiments (details in Section 5).

---

[1]The predictor $P_\theta$ can also accept a state history, $s_{t-h:t}$ where $h$ is the number of previous timesteps.

***Illustrative Example.*** *Consider Figure 2, where the same robot (blue car) is deployed in two different contexts: one where the robot drives behind a stopped truck ($C_1$) and the other where it navigates an intersection ($C_2$). The robot's reward is a tuned linear combination of control effort, collision penalty, road progress, and lane-keeping.*

*In scenario $C_1$, the robot mispredicts that the stopped truck will accelerate; the robot initiates a lane change and collides. In scenario $C_2$, the robot aggressively swerves to avoid nearby traffic. Because of the absolute reward difference between the best and the executed action (right column, Figure 2), canonical regret regards both scenarios as equally poor: $C_1$ has a regret of $11.4$ in $C_2$ has $11.7$. While the robot could have taken a more optimal action in both scenarios, canonical regret is unable to identify that the robot's decision in $C_1$ is more severely suboptimal than the decision in $C_2$.*

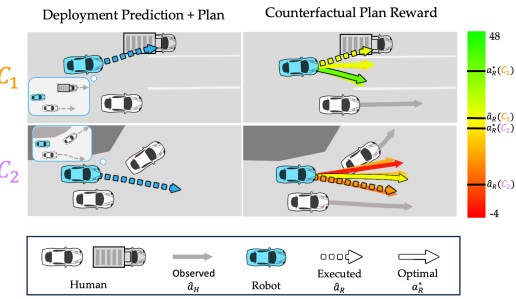

Figure 2: **Illustrative Example.** Left column: robot's predictions and deployment-time decision. Right column: the counterfactual analysis given observed human behavior. Each hindsight-optimal action and executed action's reward is on the right.

**A Generalized Regret Metric.** To address this calibration issue we propose that instead of looking at the *absolute reward* of a decision, we consider the *likelihood* of it. This probabilistic interpretation normalizes the quality of a decision relative to the context-dependent behavior distribution and is a principled way to place all decision comparisons on the same scale—a value between zero and one.

Specifically, let $\mathbb{P}_\phi(\boldsymbol{a}^{\mathrm{R}} \mid \hat{\boldsymbol{a}}^{\mathrm{H}_1:\mathrm{H}_M}, \hat{\boldsymbol{s}}, C)$ be a likelihood model that yields the counterfactual probability of any candidate robot action sequence ($\boldsymbol{a}^{\mathrm{R}}$) given the observed actions of the other human agents ($\hat{\boldsymbol{a}}^{\mathrm{H}_1:\mathrm{H}_M}$) and the observed joint state trajectory ($\hat{\boldsymbol{s}}$) in that context $C$. Constructing this likelihood model only requires access to the robot's planner; hence, it shares the same parameters as the robot used for decision-making ($\phi$). For example, if the robot is using a reward-based planner, then this probability distribution is shaped according to the same reward weights and features that the robot used for decision-making (see example in Equation 4). Additionally, this generalization to probability-space allows us to calculate regret for generative neural planners, where the likelihood model shares the same weights as the neural planner. For any human-robot deployment interaction $d := (\hat{\boldsymbol{s}}, \hat{\boldsymbol{a}}^{\mathrm{R}}, \hat{\boldsymbol{a}}^{\mathrm{H}_1:\mathrm{H}_M}) \in \mathcal{D}$, we quantify the **generalized regret** of a robot's action at timestep $t$ as:

$$\mathrm{Reg}_t(d) := \max_{\boldsymbol{a}_t^{\mathrm{R}}} \mathbb{P}_\phi(\boldsymbol{a}_t^{\mathrm{R}} \mid \hat{\boldsymbol{a}}_t^{\mathrm{H}_1:\mathrm{H}_M}, \hat{\boldsymbol{s}}, C) - \mathbb{P}_\phi(\hat{\boldsymbol{a}}_t^{\mathrm{R}} \mid \hat{\boldsymbol{a}}_t^{\mathrm{H}_1:\mathrm{H}_M}, \hat{\boldsymbol{s}}, C). \tag{3}$$

***Illustrative Example.*** *Consider the reward-based planning example from Figure 2. We model $\mathbb{P}_\phi$ using the Luce-Shepard choice rule [44], which places exponentially more probability on robot decisions that are high-reward. Our generalized regret metric results in an intuitive failure separation where canonical regret failed: $C_1$ has a higher regret of $0.56$ compared to $0.34$ in $C_2$.*

## 5  Simulation & Hardware Experimental Setup

We first instantiate our metric in the autonomous driving setting. Our prediction model $P_\theta$ is an ego-conditioned Agentformer model [45] which takes as input a candidate robot (i.e., ego) trajectory $\boldsymbol{a}_t^{\mathrm{R}}$, a history of all the vehicles in the scene $s_{t-h:t}$, and the map information $C$ to output a prediction of the behavior of all human (i.e., non-ego) vehicles $\boldsymbol{a}_t^{\mathrm{H}_1:\mathrm{H}_M}$. Our planner $\pi_\phi$ is an off-the-shelf reward-based MPC planner [46] who's parameters $\phi$ are weights of a hand-tuned reward function consisting of lane progress, lane-keeping, collision cost, and control cost (see Appendix 9.1). We pretrain $P_\theta$ on the nuScenes training split [47] and we obtain our closed-loop deployment dataset $\mathcal{D}$ consisting of 96 simulated 20-second scenarios from the BITS simulator [48] where each scenario's map and initial agent history are initialized from the nuScenes validation split.

We further instantiate our approach for generative robot planners on an Interbotix LoCoBot [49], shown in Figure 3. The LoCoBot has an Intel D435 RGB-D camera and RPLIDAR A2 2-D Lidar

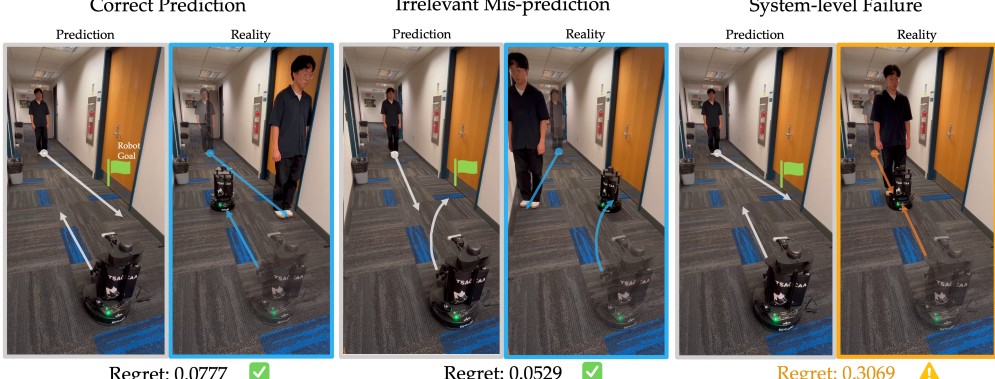

| Correct Prediction | Irrelevant Mis-prediction | System-level Failure |
|---|---|---|
| Prediction  Reality | Prediction  Reality | Prediction  Reality |
| Regret: 0.0777 ✅ | Regret: 0.0529 ✅ | Regret: 0.3069 ⚠️ |

Figure 3: Left: The robot correctly predicts the human will block its goal and proceeds straight. Middle: The robot incorrectly predicts that the human will walk straight ahead, however it is able to reach its goal despite the misprediction. Right: The robot mispredicts that the human will block its goal and proceeds straight, colliding with the human who also walked straight ahead. The robot's executed trajectory is unlikely conditioned on the human's true actions and is assigned high regret.

mounted on a Kobuki mobile base. We model the robot as a Dubins' car [50] which must reach a goal position $g^R$ unless it would cause a collision with a nearby freely moving pedestrian, in which case it navigates to a backup location (Figure 3). For constructing the robot's generative planner $\pi_\phi$, we model the joint human-robot state as the states of both agents along with a low-dimensional behavior cue from the human $\delta^H$ (e.g., eye-gaze, head tilt) that leaks information about the human's future trajectory. We take $s \equiv \delta^H$ and fix all other initial conditions between deployments. The context of the scene is the intended goal of the robot $C \equiv g^R$. The generative planner $\pi_\phi$ is an encoder-decoder architecture based on the vector-quantized variational autoencoder (VQ-VAE). It is trained with a dataset of 10,000 simulated joint human-robot trajectories $(a^R, a^H)$ generated via simple rules (described in Appendix). The encoder $P_\alpha(z \mid \delta^H, g^R)$, with learnable parameters $\alpha$, encodes the human observation $\delta^H$ and the original goal $g^R$ and produces a categorical distribution over latent embedding through vector quantization. The decoder $P_\beta(a^R, a^H \mid z)$, with learnable parameters $\beta$, takes the latent vector $z \sim P_\alpha(\cdot \mid \delta^H, g^R)$ and approximates the joint distribution over actions $a^R, a^H$. Implementation details on datasets and hyperparameters are included in Appendix 9.2.

**Generalized Regret Computation: Reward-based Planner.** Our likelihood model follows from the Luce-Shepard choice rule [44] where the parameters $\phi$ denote the weights of the reward function:

$$\mathbb{P}_\phi(a^R \mid \hat{a}^{H_1:H_M}, \hat{s}, C) = \eta \exp\{\mathcal{R}_\phi(a^R, \hat{a}^{H_1:H_M}, \hat{s}, C)\}, \tag{4}$$

where $\eta := 1/\sum_{\bar{a}^R} \exp\{\mathcal{R}_\phi(\bar{a}^R, \hat{a}^{H_1:H_M}, \hat{s}, C)\}$. For each timestep in deployment data $d \in \mathcal{D}$ where the robot originally re-planned, we re-compute the reward of each candidate ego action with respect to the ground truth behavior of the other agents, instead of predictions. With this we compute the hindsight likelihood of each candidate ego action and obtain the scene's regret: $\sum_t \text{Reg}_t(d)$.

**Generalized Regret Computation: Generative Planner.** In our hardware experiments, the human's positions were recorded as $(x, y)$ planar positions at 12 Hz using the robot's LIDAR, and transformed to a global coordinate frame of a precomputed SLAM map. The behavioral cue $\delta^H$ is obtained by tracking the position of the human and calculating the angle between timesteps, which is given to the planner to generate a 6-timestep action trajectory $\hat{a}^R$ which is executed at 2 Hz. After each action is executed, we recorded smoothed measurements of the human's trajectory by taking a simple moving average of the human position and measuring the angles between subsequent positions to obtain $\hat{a}^H$. We construct the likelihood model using Bayes' theorem where we first form a kernel density estimate of $\mathbb{P}_\phi(\hat{a}^R \mid \hat{a}^H, \delta^H, g^R)$ by sampling from the the planner 250 times and approximate the conditional probability of $\hat{a}^R, \hat{a}^H$ by integrating the KDE around a small neighborhood of the ground-truth joint action trajectory. The conditional probability of $\hat{a}^H$ is approximated by marginalizing over the space of possible $a^R$. Leveraging the approximate probabilities, we com-

pute an estimate of the counterfactual probability for any $\hat{\boldsymbol{a}}^{\mathrm{R}}$:

$$\mathbb{P}_\phi(\hat{\boldsymbol{a}}^{\mathrm{R}} \mid \hat{\boldsymbol{a}}^{\mathrm{H}}, \delta^{\mathrm{H}}, g^{\mathrm{R}}) = \frac{\sum_z P_\beta(\hat{\boldsymbol{a}}^{\mathrm{R}}, \hat{\boldsymbol{a}}^{\mathrm{H}} \mid z) P_\alpha(z \mid \delta^{\mathrm{H}}, g^{\mathrm{R}})}{\int_{\bar{\boldsymbol{a}}^{\mathrm{R}}} \sum_z P_\beta(\hat{\boldsymbol{a}}^{\mathrm{H}}, \bar{\boldsymbol{a}}^{\mathrm{R}}, \mid z) P_\alpha(z \mid \delta^{\mathrm{H}}, g^{\mathrm{R}})}.$$

This is a particular instantiation of $\mathbb{P}_\phi$ where the planner and predictor both share parameters (i.e., $\phi \equiv \theta$) given by $\alpha$ and $\beta$. We repeat this process to evaluate the maximum probability action trajectory[2], giving us both terms necessary for our generalized regret computation $\mathrm{Reg}_t(\cdot)$ in Equation 3.

## 6 Experimental Results: Detecting System-Level Prediction Failures

### 6.1 How does Regret Compare to Other Prediction Failure Metrics?

We first quantitatively and qualitatively compare scenarios extracted by our generalized regret metric (**GRM**) against three alternatives: average displacement error (**ADE**), a component-level metric that measures the $L_2$ error between the true and predicted trajectory, the system-level metric of Farid et al. [37] (**TRFD**), and the canonical variant of our system-level regret metric (**RM**). We apply each metric to the initial closed-loop simulated driving deployment data as described in Section 5. For **GRM**, **ADE**, and **RM**, we construct the failure dataset as the top 20-quantile scenes[3] (i.e., 20 scenarios with 20-second long interactions) as scored by the metric, yielding $\mathcal{D}_{\mathcal{F}}^{\mathbf{GRM}}, \mathcal{D}_{\mathcal{F}}^{\mathbf{ADE}}$ and $\mathcal{D}_{\mathcal{F}}^{\mathbf{RM}}$. **TRFD** is unique in that it only assigns *binary labels* if the scenario is a system-level failure. The pairwise overlap between $\mathcal{D}_{\mathcal{F}}$ from each metric is visualized in Figure 4.

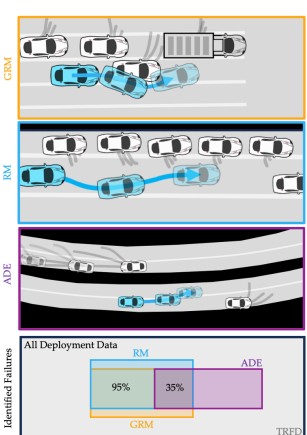

Figure 4: Qualitative comparison between scenarios uniquely identified by each metric.

**ADE** is a component-level metric and has only 35% overlap with scenes identified by the two regret metrics (**GRM**, **RM**). Qualitatively, the scenarios with high displacement error tended to have many agents or a single agent that the baseline Agentformer model largely mispredicts. An example of a scenario with high ADE but not flagged by our two regret metrics is shown in Figure 4. The ADE in this scenario is dominated by the robot consistently mispredicting a faraway vehicle to change lanes. This indicates that component-level ADE may attend to irrelevant prediction failures instead of system-level failures.

**TRFD** is a system-level metric that flags an interaction as a failure if the incurred robot reward is in the bottom $p$-quantile of the robot's reward distribution predicted during planning. Since this metric relies on a *predicted* reward distribution, it is fundamentally linked to the quality of the robot's predictions of other agents' behavior. We find that TRFD assigns *every scenario* as being anomalous (Figure 4). We hypothesize the distribution shift between the nuScenes data (that the baseline Agentformer model was trained on) and the simulated human behavior of the BITS agents causes the predicted reward distribution to be extremely misaligned with the realized reward, causing all interactions to be flagged as failures. Instead of comparing the realized outcomes with the predicted reward distribution, **GRM** and **RM** compares against counterfactual ground-truth information, making them agnostic to the base predictor quality.

**RM** is our absolute-reward variant of likelihood-based regret metric, **GRM**. Because they are both grounded in the idea of regret, **RM** has 95% overlap with **GRM**. The scenario uniquely identified by each metric is shown in Figure 4. Notably, **GRM** identifies an additional scenario where the ego vehicle drives through stopped traffic (top, Figure 4), whereas **RM** identifies an instance of the robot driving near unmoving vehicles and showing slightly conservative behavior (middle, Figure 4). Although downstream rewards are a common evaluation criteria for task-aware trajectory forecasting [37, 38], these results show that reward alone suffers from calibration challenges and may overlook pertinent system-level failures.

### 6.2 Can We Detect System-Level Failures for Reward-Free Generative Planners?

---

[2]We find this maximizing $\boldsymbol{a}^{\mathrm{R}}$ by searching over a prespecified set of action trajectories.

[3]The choice of $p$ is a design decision. We chose $p = 20$ due to the small size of the deployment dataset.

The qualitative results of our **GRM** approach in the real-world generative planning setting are shown in Figure 3. Importantly, no other system-level metrics (**TRFD**, **RM**) generalize to generative planners because they require explicit reward functions. We find that our approach only rates interactions as high-regret (0.307 on a scale from 0 to 1) when mispredictions lead to unexpectedly close proximity interactions between the robot and human (right, Figure 3). In both nominal interactions (left, Figure 3) and interactions where mispredictions are irrelevant for robot performance (middle, Figure 3), the deployment regret is close to zero (0.078 and 0.053). Overall, our new regret metric captures our motivating intuition on system-level prediction failures without the need for manually specifying an evaluative reward-function for either the high-level or low-level robot and human behaviors.

| Fine-tuning Data | ADE / FDE ($\downarrow$) | | |
|---|---|---|---|
| | nuScenes | High-Regret | Low-Regret |
| None (Base model) | 0.371 / 1.919 | 0.639 / 2.999 | 0.610 / 2.375 |
| Low-regret-FT | 0.353 / 1.554 | 0.386 / 2.333 | 0.386 / 1.783 |
| Random-FT | **0.315 / 1.435** | 0.366 / 2.100 | 0.382 / 1.738 |
| High-regret-FT | 0.327 / 1.520 | **0.333 / 2.014** | **0.366 / 1.713** |
| All-FT | *0.297 / 1.381* | *0.326 / 1.973* | *0.340 / 1.665* |

Table 1: **Fine-tuned Open-loop Prediction Performance.** Average ADE / FDE of the base $P_\theta$ and the fine-tuned predictors on log-replay data. Columns are different log-replay validation data splits: original nuScenes data, closed-loop simulation data that does exhibit system-level failures (i.e. high-regret) and doesn't (i.e., low-regret). Lowest error denoted in italics-bold, and second lowest in bold. Each baseline was trained with three random seeds and the reported numbers are from best-performing model.

## 7 Case Study: Mitigating Prediction Failures via High-Regret Fine-Tuning

We demonstrate one potential use for system-level failure data by fine-tuning the Agentformer model on high-regret interactions. We first hold out 20 (comprised of 17 low-regret and 3 high-regret scenarios) of the 96 deployment scenarios for evaluation and leave the rest (76) as potential data for fine-tuning. We compare the closed-loop simulation (Table 2) and open-loop prediction (Table 1) performance of Agentformer fine-tuned on four different data subsets. **Low-regret-FT** uses the 20 scenarios with the lowest regret, **Random-FT** uses 20 random scenarios from the 76 scenarios available for fine-tuning, **High-regret-FT** uses the 20 scenarios with the highest regret, and **All-FT** is a privileged baseline with access to all 76 scenarios. During fine-tuning we unfreeze only the final 1.5M parameters of the 17.7M parameter Agentformer model, and we examine the results from fine-tuning the model on each dataset for 3 random seeds. Details and ablations in Appendix 10.

*Takeaway 1: **High-Regret-FT** improves log-replay prediction accuracy in high-regret scenarios, without degrading performance on low-regret scenarios and the original nuScenes validation set.*

We first evaluate the predictors "open-loop" on log-replay data from three different validation datasets: the original nuScenes validation data, the 3 high-regret scenes in the deployment validation dataset, and the 17 low-regret scenes in the deployment validation dataset. We report the Average Displacement Error (ADE) and Final Displacement Error (FDE) for each baseline's best performing seed in Table 1. The base predictor doubles its ADE and FDE on the simulated deployment data due to a distribution shift in the driving behavior between nuScenes and simulation. Intuitively, **High-Regret-FT** has the most ADE and FDE improvement on high-regret validation data. We hypothesize that the improved open-loop prediction accuracy on low-regret scenes is because the high-regret validation data leaks enough information about the general distribution shift between the nuScenes and the BITS simulation to enable improvements on nominal interaction data as well. While **All-FT** achieves the best prediction accuracy across the board, it is important to note that this is an information-advantaged baseline: it has the ability to learn from both system-level and non-system-level failures the deployment data. However, **High-Regret-FT** is much more data efficient to train, rivaling the performance of **All-FT** despite using 77% less data. As the deployment dataset $\mathcal{D}$ grows and training time and costs increase along with it, this can be a promising avenue to alleviate these challenges while still achieving high predictive performance. Finally, the least improvement over the base model is exhibited by **Low-Regret-FT** and followed by **Random-FT** which demonstrates the quantitative value of learning from high-regret data.

*Takeaway 2: Including high-regret data during fine-tuning improves closed-loop performance.*

Our first evaluation metric for measuring closed-loop re-deployment performance differences is *Total Collision Cost*, which is the total cost the ego incurred from collisions. Second, to disentangle severe but infrequent collisions from frequent clipping / grazing we measure *Collision Severity*=$\frac{\text{Total Collision Cost}}{\text{\#Frames in Collision}}$. Finally, we measure average *Regret* as defined in Section 4. We report average closed-loop performance for each method across three seeds of fine-tuning in Table 2.

| Fine-tuning Data | High-Regret Scenarios | | | Low-Regret Scenarios | | |
|---|---|---|---|---|---|---|
| | Col Cost (↓) | Col Severity (↓) | Regret (↓) | Col Cost (↓) | Col Severity (↓) | Regret (↓) |
| Base Model | 10.829 | 0.848 | 0.034 | **0.411** | *0.051* | **0.006** |
| Low-regret-FT | 7.489 (± 0.408) | 0.794 (± 0.059) | 0.019 (± 0.002) | 0.670 (± 0.206) | 0.354 (± 0.493) | 0.009 (± 0.000) |
| Random-FT | 7.209 (± 0.883) | 0.743 (± 0.064) | **0.016 (± 0.003)** | 0.495 (± 0.213) | 0.076 (± 0.054) | **0.006 (± 0.000)** |
| High-regret-FT | **3.763 (± 0.965)** | **0.588 (± 0.101)** | **0.016 (± 0.003)** | *0.397 (± 0.814)* | 0.062 (± 0.031) | **0.006 (± 0.001)** |
| All-FT | *1.176 (± 0.775)* | *0.377 (± 0.067)* | *0.013 (± 0.002 )* | 0.486 (± 0.131) | **0.060 (± 0.030)** | 0.007 (± 0.001) |

Table 2: **Closed-Loop Re-Deployment Performance.** Average robot performance metrics for each fine-tuned predictor (standard deviation over three seeds of fine-tuning in parentheses). Two main columns are evaluations on high-regret and low-regret held-out validation scenes. Italics-bold indicates best, and bold indicates second best. Across all metrics and settings, High-regret-FT consistently improves robot performance, and is competitive with the privileged All-FT approach.

We find that **High-Regret-FT** decreases collision cost, severity, and average regret compared to the base model across both high- and low-regret scenes. As hypothesized, most closed-loop performance gains are observed in the high-regret scenes, reducing robot regret by $-53\%$ and collision cost by $-65\%$ (shown in Table 2). By being able to see all the deployment data, **All-FT** observes the highest performance gains in high-regret scenes but shows slight (but not statistically significant) performance degradation in low-regret scenes. For each baseline, the degradations from the base model are within 1 standard deviation (except for **Low-Regret-FT**). In contrast, the improvement of **High-Regret-FT** is more than one standard deviation better than the performance of **Low-Regret-FT** and **Random-FT** with respect to the collision-based metrics. The closed-loop performance gains resulting from fine-tuning on high-regret data is shown qualitatively in Figure 5.

*Takeaway 3: Closed-loop and open-loop performance are only correlated on high-regret scenes.*

We find that in the high-regret scenes, the open-loop performance (Table 1) is positively correlated with the closed-loop planner performance (Table 2). We hypothesize that this is because these scenes are much more challenging (e.g., highly interactive), encode more nuanced human behavior, and affect the planner the most; thus, the prediction improvements in such scenarios indicate that $P_\theta$ has learned to predict chal-

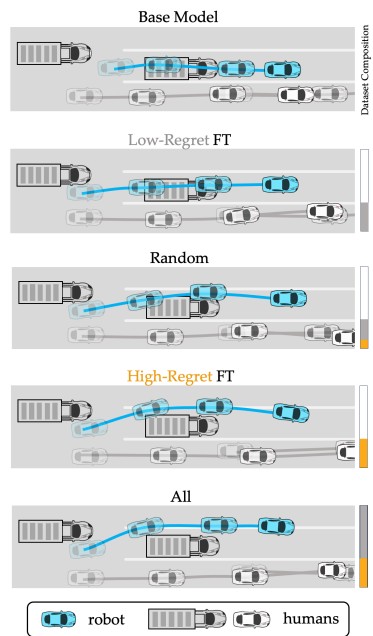

Figure 5: As $P_\theta$ is fine-tuned on more high-regret interactions, closed-loop performance improves.

lenging interactions better, naturally improving the closed-loop planning performance.

# 8   Limitations & Future Work

One limitation of our system-level failure detection method is the need for a designer to set $p$ when detecting scenarios with the top $p$-quantile regret. Poorly tuning this $p$ could result in classifying benign scenarios as anomalous or overlooking true system-level failures. In the generative planner setting, extremely out-of-distribution interaction data (e.g., not in the support of the training distribution) could lead to poor regret estimates. To remedy this, we hypothesize that our approach can be complemented by an anomaly detector that can detect highly out-of-distribution data during deployment. Furthermore, our hardware experiments underscored how the logged data collected by the robot during deployment (e.g., states of agents) must be sufficiently high quality to compute accurate regret. Failures in detection/tracking from perception modules in real-world deployments could corrupt the data used to calculate regret and misidentify system-level failures.

While we instantiated our approach in simulated and hardware robot navigation interactions, collaborative manipulation [4]) is an exciting future direction. Finally, future work should also investigate alternative uses for the datasets constructed with our regret metric, e.g., to inform runtime monitors that can anticipate system-level prediction failures online.

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

# 9 Experiment Implementation Details

## 9.1 Reward-based Planner & Generative Predictor

**Robot Policy: Tree Policy Planning (TPP).** Our MPC planner $\pi_\phi$ is a tree-structured contingency planner developed by Chen et al. [46]. This reward-based planner is compatible with state-of-the-art conditional behavior prediction models to account for the influence of the robot's actions when planning. The planner first generates a tree of randomly sampled multi-stage dynamically feasible trajectories, after which each branch is fed into an ego-conditioned predictor $P_\theta$ to generate a scenario tree comprised of multimodal predictions. The optimal robot trajectory is obtained by maximizing a reward function which in our experiments consisted of a linear combination of collision costs, lanekeeping costs, control costs, and distance-traveled reward (full details can be found in the original paper [46]). This reward function is fixed throughout the entire robot deployment and human predictor fine-tuning process.

**Base Human Predictor.** The training loss for our ego-conditioned Agentformer model is a weighted sum of three terms: 1) prediction loss that penalizes incorrect predictions from the ground truth, 2) ego-conditioned collision (EC) loss that penalizes collisions between the predictions and conditioned ego-action. 3) other regularization losses such as diversity and KL divergence loss. Our base prediction model is trained on the Nuscenes dataset [47], consisting of 1,000 20-second driving logs collected in Boston and Singapore. During the initial training of the predictor we used a learning rate of 1e-4.

**Dataset Construction.** Our deployment dataset $\mathcal{D}$, consisted of $N = 96$ total closed-loop deployment scenarios each of length 20 seconds. We ranked the ego agent's average regret and chose the top $p = 20$-quantile of scenarios to be our set of scenes that suffered from system-level prediction failures, yielding a total of 20 high-regret scenarios in $\mathcal{D}_\mathcal{F}$. The rest of the 76 scenarios were deemed sufficiently low-regret. We hold out 20% of both high-regret $\mathcal{D}_\mathcal{F}$ and low-regret $\mathcal{D} \setminus \mathcal{D}_\mathcal{F}$ scenes (3 and 17 respectively). The remainder of the deployment data is used for fine-tuning the model.

## 9.2 Generative VQ-VAE Planner

**Dataset Generation.** We constructed a dataset of $|\mathcal{D}_{train}| = 10,000$ human-robot trajectories based on a small set of simple rules in order to train our generative planner. The positions of the two goal locations and the robot and human's start locations were fixed in all training examples (shown in Figure 3), but the choice of goal $g$ and initial human behavior cue $\delta_\mathrm{H}$ were uniformly distributed. The synthetic human trajectory generation abides by the following rule: $a^\mathrm{H} = PC(\texttt{left})$ if $\delta_H + \epsilon < 1/3$, $PC(\texttt{right})$ if $\delta_\mathrm{H} + \epsilon > 2/3$, $PC(\texttt{straight})$ otherwise. Here, $PC(\cdot)$ denotes a proportional controller noisily executing a 6-timestep action trajectory that guides the human to the prespecified goal points to the left, right, or straight-ahead from the human's initial position, and $\epsilon$ is zero-mean noise.

The synthetic robot trajectory generation follows a similar process, but conditions on the desired goal $g_R$ and the synthetic human behavior $a^\mathrm{H}$. The robot's action $a^\mathrm{R}$ is a noisily executed 6-timestep action trajectory given by a proportional controller that guides the robot to $g_R$, unless this brings the robot in close proximity to the human trajectory $a^\mathrm{H}$ in which case the robot's proportional controller guides it to the other goal with 80% probability. This dataset synthesis leads to six different high-level outcomes of the interaction depending on which directions the human and robot travel. Figure 3 shows three of these joint human-robot interaction outcomes.

**Generative Planner Architecture.** The robot's generative planner $\pi_\phi$ used an encoder-decoder architecture based on the VQ-VAE, which is a variant of the traditional VAE that uses vector quantization to discretize the latent representation into $K$ latent embedding vectors $\{z_k\}_{k=1}^K$. The encoder $P_\alpha(z \mid \delta_\mathrm{H}, g)$ with learnable parameters $\alpha$ takes $\delta_\mathrm{H}$ and $g_R$ as inputs to encode the input and produce a categorical distribution over latent embedding through vector quantization. We further inject a small amount of zero-mean Gaussian noise into quantized embedding $z \in \{z_k\}_{k=1}^K$ to encourage diverse outputs during the joint trajectory decoding phase. This decoder $P_\beta(a^\mathrm{R}, a^\mathrm{H} \mid z)$ with learnable parameters $\beta$ takes a sampled latent vector $z$ as input (sampled from $P_\alpha(z \mid \delta_\mathrm{H}, g)$ with additional noise injection) to approximate the joint distribution over actions $a^\mathrm{R}, a^\mathrm{H}$. In our setting,

we set $K = 6$ and initialized the latent embeddings with K-means clustering to encourage each of the K embedding vectors to correspond to one of the six joint human-robot action trajectories.

## 9.3 Hardware Experiments

The hardware experiments were conducted on an Interbotix LoCoBot [49] equipped with an Intel RealSense D435 RGB-D camera and RPLIDAR A2 360 degree LIDAR. Offline, the robot generated a occupancy map of the environment with SLAM from RGB-D observations. This SLAM map was used to generate a global coordinate frame for the human and robot. The human's position with 2-D LIDAR scan data and filtering out LIDAR measurements from static objects such as walls. The remaining LIDAR sensor measurements were then averaged to generate a noisy $(x, y)$ points of the human's position relative to the robot that were measured at 12 Hz. These positions were then translated to the global coordinate frame obtained by transforming from the robot's local coordinate frame to the global coordinate frame.

We modeled the human as a single integrator with constant velocity where the control action was a heading direction. These were extracted by taking a simple moving average of the position measurements to obtain a position measurements with reduced noise at a rate of 2 Hz. The actions at each time step were taken to be the angle between the current and previous human position measurement in the global coordinate frame.

## 10 Fine-tuning Details & Ablations

| Fine-tuning Data | High-Regret Scenarios | | | Low-Regret Scenarios | | |
|---|---|---|---|---|---|---|
| | Col Cost | Col Severity | Regret | Col Cost | Col Severity | Regret |
| Base Model | 10.829 | 0.848 | 0.034 | 0.411 | 0.051 | 0.006 |
| AvgReg | **4.286 (-60.4%)** | 0.690 (-18.7%) | 0.014 (-58.7%) | *0.347 (-15.5%)* | *0.039 (-24.9%)* | *0.005 (-11.1%)* |
| WstReg | 4.713 (-56.6%) | **0.520 (-38.7%)** | 0.020 (-43.3%) | 0.565 (+37.4%) | 0.093 (+80.9%) | 0.006 (+2.02%) |
| AvgReg (+col, +div) | 14.505 (+34.0 %) | 0.960 (+13.2 %) | 0.048 (+39.4 %) | 0.437 (+6.4 %) | 0.082 (+60.3 %) | 0.006 (+9.1 %) |
| AvgReg (+div) | 7.477 (-30.9 %) | 0.654 (-23.0 %) | **0.014 (-60.6 %)** | 0.366 (-10.83 %) | 0.0578 (+12.4 %) | 0.005 (-6.06 %) |

Table 3: Finetuning when ranking with average regret (2nd row, also reported in Table 2 as High-regret-FT) vs worst single-timestep regret (3rd row). We also ablated the choice of loss function on closed-loop performance by using all of the original loss functions (4th row) and only removing the ego-conditioned collision loss (5th row).

**Fine-tuning the Predictor.** For fine-tuning, we unfreeze only the last 1.5M parameters of the 17.7M parameter Agentformer model. This corresponds to the last half of the future decoder module [45]. Across all our baselines, we fine-tune the predictor to see the data for 25 epochs at a reduced learning rate of 5e-5 held constant throughout the 25 epochs. Furthermore, when fine-tuning we eliminate the diversity loss and EC collision loss, focusing the model's efforts to only accurately predict the ground-truth deployment data. We ablate this choice in 3 on one seed. Due to the large size of the Nuscenes data, we trained on two Nvidia A6000 GPUS with a batchsize of 14 (effective batch size 28) and gradient accumulation of 5 steps.

**Regret Formulation and Mitigation Methods.** In our formalism from Section 4, we defined the regret on a per-timestep basis. For the experiments, we ranked the scenarios by the average regret incurred over the 20-second interaction horizon $\hat{T}$. However, inspired by the safety analysis literature that considers only the *worst* safety violation incurred over the time horizon [51], we also experimented with ranking scenarios by the *worst* single-timestep regret incurred over $\hat{T}$. The closed-loop simulation results are shown in the second row of Table 3.

Furthermore, during predictor fine-tuning described in Section 5, we chose to remove the ego-conditioned collision loss and diversity loss during the fine-tuning. We found that when fine-tuning with the ego-conditioned (EC) collision loss, the predictor was unable to learn from the high-regret data. This is because much of the high-regret data was induced by ego collisions with other agents. Thus, the EC collision loss directly conflicts with learning from this new data. The results of fine-tuning with EC collision loss and diversity loss are shown in the third row of Table 3. We also observed that training without diversity loss leads to slightly better closed-loop performance. The baseline fine-tuning without EC collision loss but with diversity loss is shown in the fourth row of Table 3.

# 11    Case Study: System-level Perception Failure Detection via Regret

We demonstrate a toy setting where our regret metric is able to identify task-relevant *detection* failures when using a Conditional Variational Autoencoder (CVAE)-based robot navigation planner.

**Problem Setup.** A robot is deployed to reach a goal point $g_1$ (top flag in Figure 6). It has dynamics $\dot{\mathbf{x}} = [\dot{x}, \dot{y}, \dot{\theta}]^\top = [\cos\theta, \sin\theta, u]^\top$, where $u$ is the control input. It is also equipped with a sensor $P_{sense}(C)$ that returns $1$ if it detects an obstacle $\mathcal{O}$, and $0$ otherwise. In the absence of an obstacle, it should always try to reach $g_1$. However, if an obstacle is blocking the robot's path to $g_1$, it should instead navigate to an alternate goal location $g_2$ (right flag in Figure 6).

To train our robot planner, we generated a dataset of $|\mathcal{D}_{train}| = 10000$ trajectories. For simplicity, the robot's starting state and the locations of $g_1$, $g_2$, and $\mathcal{O}$ (if present) were the same across the entire training dataset. In each scene, the demonstrated trajectories were obtained by applying a small amount of Gaussian noise to a proportional controller guiding the robot to its goal location.

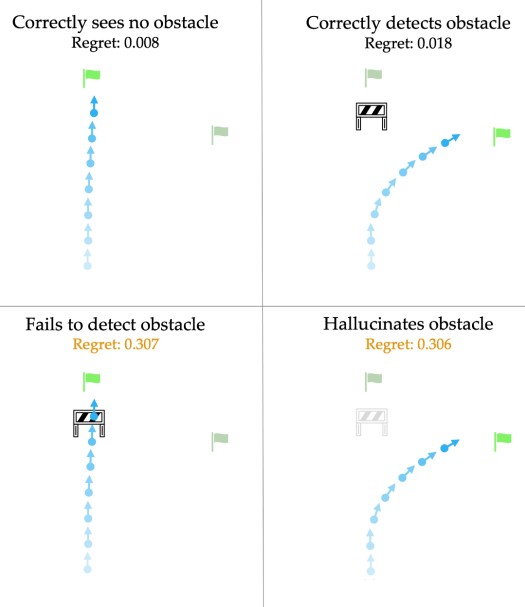

Figure 6: **Generative Planner: Perception Failure.** The robot's training data demonstrated that the robot should aim for the top flag in the absence of an obstacle, but move toward the rightmost flag if an obstacle is detected. The green flag denotes the robot's perception-conditioned planned goal. Orange obstacle denotes that there was actually an obstacle present, and grey obstacle denotes that the robot falsely detected an obstacle. Our regret metric correctly flags detection failures and measured nominal scenarios as having low regret.

**Inducing Generative Planner Failures.** We wanted to identify which robot behaviors (if any) were assigned high regret by our regret metric. To do this, we deployed the robot four times. In two of the scenarios, there was the obstacle $\mathcal{O}$ (top right, and bottom left of Figure 6). However, in one of those two scenarios, we injected a perception failure (i.e., manually setting the sensor observation to be incorrect) that prevented the robot from detecting $\mathcal{O}$. We similarly deployed the robot in a setting with no obstacle two times but injected a perception failure that falsely detected $\mathcal{O}$ in one of the deployments (bottom right in Figure 6).

**Trajectory Probabilities.** To simulate the robot, we sampled 500 trajectories conditioning on $P_{sense}(C) = 1$ and 500 on $P_{sense}(C) = 0$. For each condition, we took the average of the 500 respective trajectories as an exemplar for deployment. However, despite the ease of sampling from the CVAE, finding the probability of a particular sample from a generative model is known to be a difficult problem [52]. To approximate each exemplar trajectory's probability under the ground truth presence/absence of the obstacle, we formed a per-timestep kernel density estimate (KDE) of the conditional likelihood using the 1000 previously sampled trajectories (For more details, see Fig-

ure 5 of [53]). To compute the trajectory probability, we integrated the KDE for $\pm\delta$ around the most likely and executed action for $\delta = 0.1$ at each timestep. The trajectory's regret was computed via our metric as in Equation 3 by averaging over the per-timestep regret.

**Results.** As seen in Figure 6, the robot incurs very low regret whenever its sensor was accurate. However, in both settings where the sensor failed, the robot incurs a much higher regret. We highlight that nowhere in this formulation (demonstrations nor generative planner) was there a reward function that determined the robot's behavior. Furthermore, this toy example demonstrates that our general calibrated regret metric may have use beyond the human-robot interaction setting; for example, for detecting and mitigating failures in other autonomy modules beyond agent prediction (e.g., perception module) in a task-aware manner [34].

