# OpenReview forum: "Not All Errors Are Made Equal: A Regret Metric for Detecting System-level Trajectory Prediction Failures"
_robot-learning.org/CoRL/2024/Conference — CoRL 2024_

### Official Review · Reviewer_ehM7 · 2024-07-19
**Review of "Not All Errors Are Made Equal: A Regret Metric for Detecting System-level Trajectory Prediction Failures"**

**Originality:** 3
**Technical Quality:** 3
**Clarity Of Presentation:** 3
**Potential Impact:** 3
**Recommendation:** 1
**Confidence:** 4

**Review:**

The paper introduces a unique approach to identifying impactful prediction failures using the concept of regret, marking a significant theoretical contribution. Demonstrating the method in the practical setting of autonomous driving showcases its real-world applicability. It highlights that fine-tuning models on high-regret data, which constitutes a smaller portion of the dataset, can achieve performance improvements comparable to using the entire dataset. The evaluation is comprehensive, with both qualitative and quantitative comparisons to other failure detection metrics, underscoring the superiority of the proposed approach. The detailed methodology, including the experimental setup, models used, and the process of calculating generalized regret, ensures reproducibility. However, the method's requirement for setting a p-quantile for detecting failures could lead to misclassification if not well-tuned. The reliance on accurate simulation data may not always reflect real-world scenarios due to potential perception module inaccuracies. While the approach is demonstrated in autonomous driving, its applicability to other domains is not explored in-depth, and the probabilistic generalization of regret may add computational complexity, potentially limiting its real-time application. Overall, the paper presents a significant advancement in failure detection for human-robot interactions, with promising results in improving model performance efficiently. Future work should address parameter sensitivity, explore real-world applications, and extend the approach to other domains.

Cons:

•  Lack of Physical Robot Testing: The paper does not demonstrate the proposed method on physical robots. While simulations are valuable, testing on physical robots is a key expectation for CoRL submissions.

**Quality Of The Limitations Section:**

2

**Questions For Rebuttal:**

• How sensitive is the performance of the proposed metric to the choice of the p-quantile for detecting high-regret scenarios?

• Can you provide more details on how the simulation data might differ from real-world scenarios, and how this might affect the results?

• Have you considered the computational complexity of the probabilistic generalization of regret, especially in real-time applications?

**Robotics Focus:**

2

**Summary Of Paper:**

The paper proposes a novel metric based on the concept of regret to identify prediction errors that critically impact robot performance. The key contributions include formalizing system-level prediction failures, introducing a probabilistic generalization of regret, demonstrating the approach in autonomous driving simulations, and showing that fine-tuning with high-regret data can efficiently improve system performance. The results indicate that the method effectively identifies impactful failures and improves model performance with reduced data usage.

**Summary Of Recommendation:**

The paper presents a novel and effective approach to detecting system-level prediction failures using a regret-based metric. It demonstrates significant improvements in model performance with efficient data usage and offers a detailed and comprehensive evaluation. Despite some limitations regarding parameter sensitivity and real-world applicability, the contributions are substantial and valuable to the field. While simulations in CARLA or similar environments can be acceptable, the paper would be stronger if it demonstrated some level of transfer to real-world scenarios.

---

### Official Review · Reviewer_HApx · 2024-07-21
**Not All Errors Are Made Equal: A Regret Metric for Detecting System-level Trajectory Prediction Failures**

**Originality:** 2
**Technical Quality:** 3
**Clarity Of Presentation:** 3
**Potential Impact:** 3
**Recommendation:** 2
**Confidence:** 3

**Review:**

The research presented a mathematical formulation of a regret metric, with a particular focus on autonomous driving. The presented work outlines a nice contribution towards robotic systems capable of improving failure recovery. The neat mathematical formulation of this method could potentially be strong enough to have application to other safety-critical domains. The quality of the proposed method is evidenced through the mathematical formulation and simplified simulation environments. There is evidence of a successful outcome for eg through the case study. Their clarity of the methods could be improved by adding more detail on the implementation and hyperparameters (missing appendix), including github repository/documentation for replicating the experiments once the work is accepted for submission. The main strength of this work is the formulation of the problem using a regret measure (although it would be nice to differentiate more explicitly to other probabilistic introspective methods) and its main weakness is the simplified simulated environments and lack of real-world experiments.

**Quality Of The Limitations Section:**

1

**Questions For Rebuttal:**

- The method proposed by the authors has some similarity to another method illustrated in the following paper: https://ieeexplore.ieee.org/abstract/document/9561749/. I believe this should be added to the related work and the authors should outline how their method differs to this one. It would be nice if they can have some form of comparison with this method.
- In the contributions, I am not sure how you ‘show’ that fine-tuning state-of-the-art human behavior predictors with high-regret data is a promising avenue for efficiently mitigating system-level prediction failures. Could you rethink this contribution and give evaluation metrics how you illustrate this contribution through your experimental evaluation?
- In the related work you seem to touch a bit on system-level prediction failures. However, an example you give is perception which naturally your method would suffer once deployed in the real world. Could you be a bit more specific and give examples on what system-level prediction failures you would be looking at and how your method addresses these in simulation and potentially in the real world?
- I do not fully understand the motivation of using the Agentformer model. Is this the only model available or how did you conclude on this model? My understanding is that it is a very simplistic way of modelling human input.
- While I do appreciate the challenges of real-world deployment of your method and favouring using the simulator for obvious reasons, I am not sure how good your method would overcome the sim2real gap, in addition to the uncertainty with humans and other environmental noise. Could you expand a bit in your limitations how you plan to overcome these challenges? In addition, you mentioned that this work could work for collaborative manipulation - do you have any simulation results for this or more concrete plans how you have/will carry this out?
- Personally, I think the work would be hard to reproduce since you have not added any details on hyperparameters etc, although you mention section 9 and 10. My supplementary material seems to be only a video, hence I cannot assess well the reproducibility of your work. Could you update this part or link a more detailed appendix?

Minor comments:
- I found the caption for Figure 2 and the explanation for the illustrative example very repetitive. It might be worth rethinking how you present your work.

**Robotics Focus:**

2

**Summary Of Paper:**

The authors propose characterising ‘system level’ prediction failures through formulating mathematically the idea of regret. This stems from the idea that high-regret interactions are precisely those in which mispredictions caused the robot to make a suboptimal decision in hindsight. Their main area of application is autonomous driving, although I do believe their work could be impactful more broadly. The authors propose their main contributions as the following: (1) Formalise system-level prediction failures via the mathematical notion of regret; (2) Propose a probabilistic generalization of regret that is calibrated and compatible with reward-free robot planners; (3) Demonstrate their approach in closed-loop autonomous driving interactions simulated via the BITS trafficc simulator; (4) Show that fine-tuning state-of-the-art human behavior predictors with high-regret data is a promising avenue for efficiently mitigating system-level prediction failures.

**Summary Of Recommendation:**

Given the simplistic experimental evaluation, I am leaning towards revise and resubmit due to the additional experiments/details that would be needed.

---

### Official Review · Reviewer_t3uw · 2024-07-26
**Review 1**

**Originality:** 3
**Technical Quality:** 4
**Clarity Of Presentation:** 5
**Potential Impact:** 3
**Recommendation:** 4
**Confidence:** 4

**Review:**

Quality / Clarity: Overall, this is a high quality paper. It is well organized with figures that illustrate the core concepts of the paper. The introduction is compelling. The motivation and description of the method, the experiments, and the results are clear. Finally, the related work is concise but distinguishes the contribution of this paper from previous works on this topic.

Originality / Significance: The use of a probabilistic regret metric for identifying system-level failures is an original and useful idea. As such, the results are convincing and do hold significance toward increasing the robustness of mature autonomous systems in the real world.

Summary of strengths:
- The failure detection method (based on generalized regret) is general and as such is demonstrated to work for both reward-based planners and neural implicit robot planners.
- Experiments validate the downstream effectiveness of the regret metric, demonstrating an increase in closed-loop re-deployment performance after fine-tuning on classified high-regret trajectories.
- The reviewer appreciates the effort put into the organization and presentation of this paper, which makes it easy to extract the core contributions and their significance.

Summary of weaknesses:
- There are cases in which system-level failures result from never-before-seen, out-of-distribution scenarios. In these cases, it would appear that the generalized regret metric is susceptible to erroneous regret estimates (when a reward model is not available), because likelihood scores can be unreliable w.r.t. OOD inputs. Thus, a preferable robot trajectory can be assigned a low likelihood and vice versa, potentially omitting high-regret, OOD scenarios.

**Quality Of The Limitations Section:**

3

**Questions For Rebuttal:**

- Please address the weakness described above? In short, how can we trust the likelihood model to properly assess likelihoods of out-of-distribution scenarios that are likely to result in system-level failures?
- I encourage the authors to make transparent earlier on that this paper is addressing an offline failure detection problem rather than an online one.

**Robotics Focus:**

3

**Summary Of Paper:**

The paper “Not All Errors Are Made Equal: A Regret Metric for Detecting System-level Trajectory Prediction Failures” presents a framework for detecting system-level failures resulting from trajectory prediction errors in human-robot interactive scenarios (e.g., autonomous driving). The method is applied in hindsight (i.e., offline) to identify system-level failures in a large corpus of driving data. The core contribution is a generalized regret metric (GRM) that approximates regret as the difference in likelihood scores between an optimal robot trajectory and the robot trajectory actually taken. The proposed GRM 1) circumvents the need for a reward model to compute a standard regret metric and 2) provides calibrated regret scores across a wide variety of scenarios / contexts likely to be present in self-driving datasets.

**Summary Of Recommendation:**

In summary, the paper was high quality, clearly written, presented an original idea, which yielded results of significance. I therefore recommend acceptance of this paper.

---

### Author Rebuttal · Authors · 2024-08-05

We would like to thank the reviewers for their encouraging reviews and thoughtful feedback. We are delighted that all of the reviewers agree that the formulation of our generalized regret metric is novel and impactful. Based on helpful suggestions of the reviewers, **we are currently implementing a hardware demonstration of our regret metric’s ability to detect system-level failures**. These experiments will instantiate the toy generative planner example in the current manuscript and will have robot sensing fully on-board. We aim to submit the revised manuscript following the completion of the additional hardware experiment. In the meanwhile, **we have uploaded our appendix** (which was erroneously deleted during the initial submission process) for additional clarity with regards to reproducibility. We also plan on open sourcing our code after the review period if the paper is accepted.

---

### Decision · Program_Chairs · 2024-09-04

**Decision:**

Accept

**Comment:**

The manuscript received three quality reviews. The reviewers highlighted the following strengths and weaknesses, among many substantial pieces of feedback:
- (+) High-quality manuscript (well-organized text, good figures, good related work)
- (+) Well-motivated methods and good results
- (+) General approach
- (+) Good mathematical foundation
- (+) Extensive evaluation in simulation
- (-) Sensitivity to OOD cases
- (-) Make clearer that this is an offline approach
- (-) Missing hyperparameters and other implementation details, limiting reproducibility
- (-) Evaluation only on simplistic simulation
- (-) Lack of real-world experiments
- (-) Unsustained claim in contribution list
- (-) Unsustained use of a specific model (Agentformer)
- (-) Lack of a discussion on the generalization of the approach to other domains
- (-) Lack of a discussion on dependency on ground truth (simulation) data and sensitivity to the choice of the p-quantile
- (-) Lack of analysis of computation complexity and speed for real-time systems

After rebuttal and discussions among the reviewers, they agreed the quality of the paper improved and several of the concerns were covered.